# Contents of the Sexual and Reproductive Health Subject in the Undergraduate Nursing Curricula of Spanish Universities: A Cross-Sectional Study

**DOI:** 10.3390/ijerph182111472

**Published:** 2021-10-31

**Authors:** Carlos Saus-Ortega, María Luisa Ballestar-Tarín, Elena Chover-Sierra, Antonio Martínez-Sabater

**Affiliations:** 1Nursing School “La Fe”, Generalitat Valenciana, 46026 València, Spain; sausor@uv.es; 2Nursing Care and Education Research Group (GRIECE), GIUV2019-456, Nursing Department, Universitat de Valencia, 46010 València, Spain; m.luisa.ballestar@uv.es (M.L.B.-T.); antonio.martinez-sabater@uv.es (A.M.-S.); 3Nursing Department, Facultat d’Infermeria i Podologia, Universitat de València, 46010 València, Spain; 4Internal Medicine, Consorcio Hospital General Universitario de Valencia, 46014 València, Spain; 5Grupo Investigación en Cuidados (INCLIVA), Hospital Clínico Universitario de Valencia, 46010 València, Spain

**Keywords:** reproductive health, sexual health, women’s health, nursing curriculum, nursing education, undergraduate education, comprehensive sexuality education, gender equality, sexual rights

## Abstract

Background: Nursing students must receive adequate training in Sexual and Reproductive Health (SRH), which could allow them to acquire sufficient knowledge to solve the future SRH needs of everyone. In this study, the contents of the SRH subject in the undergraduate nursing curricula of 77 Spanish universities were examined to determine what SRH training nursing students are receiving. Methods: The contents of the SRH subject of all the curricula that were available online were reviewed. The distribution of the contents (topics) in the two areas (reproductive health and sexual health) was analyzed, and the prevalence of each topic was established. It was also determined whether there were differences between public (*n* = 52) and private universities (*n* = 25). Results: The training of nursing students focuses mainly on the area of Reproductive Health (15 topics). Most of the topics of this area had a prevalence greater than 50%. Although the area of Sexual Health had 14 topics, most of these topics had a low prevalence (<20%), especially in private universities. Conclusions: It was found that there is considerable variation in the distribution and prevalence of SRH topics between universities. The contents of the area of Reproductive Health are usually prevalent in most of the curricula. However, the contents of the area of Sexual Health are minimal in most of the universities. An organizational effort is required to determine and standardize the contents of SRH that nursing students should receive in Spain to avoid inequalities in their training. Guaranteeing homogeneous SRH contents will avoid deficit situations that could affect people’s care.

## 1. Introduction

Good sexual and reproductive health (SRH) is a state of complete physical, mental and social well-being in all matters relating to the reproductive system. It implies that people can have a satisfying and safe sex life, the capability to reproduce, and the freedom to decide if, when, and how often to do so. To maintain one’s sexual and reproductive health, people need access to accurate information and the safe, effective, affordable, and acceptable contraception method of their choice. They must be informed and empowered to protect themselves from sexually transmitted infections. Moreover, when they decide to have children, women must have access to services that can help them have a fit pregnancy, safe delivery, and healthy baby. Every individual has the right to make their own choices about their sexual and reproductive health [1].

Nurses are uniquely positioned to address the sexual and reproductive health needs of all people and populations at all stages of life (childhood, adolescence, adulthood, and old age) [2]. They usually interact with women based on scheduled tasks, such as cancer screening programs, gynecological concerns or problems, or difficulties related to pregnancy, childbirth, or postpartum, and with men from problems related to sexuality and reproduction, sexually transmitted infections, or other pathologies [3,4]. They also attend to the sexual and reproductive health needs of specific groups (LGBTQI+ people [5,6], communities such as the gypsy, for example, or religious or linguistic minorities, among others [7]). However, to adequately care for all people, it is essential that nurses, during their undergraduate training, receive basic SRH contents that allow them to acquire sufficient knowledge [8,9].

Cappiello, Coplon, and Carpenter, in a review of the literature, examined the extent to which the United States’ undergraduate nursing curricula include SRH contents. The authors concluded that the SRH contents taught are essential, mainly due to time constraints. These essential contents were (1) nursing care in pregnancy and prenatal health, childbirth, and postpartum care, (2) family or women’s health in the context of motherhood and parenting, and (3) disease prevention or health maintenance in the mother-child dyad and the pregnancy process. However, although they consider that this training could be improved by increasing the time to impart more content, they indicate that these essential contents are sufficient to acquire adequate knowledge and, thus, be able to provide adequate care to the women. In addition, they point out that complex clinical problems in SRH are usually addressed in postgraduate or specialized training (midwives) [5]. 

McLemore and Levi have also studied SRH subject content in the United States’ undergraduate nursing curricula. They also considered that, although the contents currently taught are essential, they guarantee the provision of comprehensive SRH care to all women and their families at all stages of life. These essential contents were (1) nursing care for disease prevention and risk reduction for women throughout life, (2) women’s health or family health in the context of preconception and normal and high-risk pregnancies, (3) crises related to women’s health and pregnancy, and (4) sociocultural, economic, political, or ethical factors that affect the provision of health services [10]. 

More recently, Simmonds et al. affirmed that SRH content in undergraduate nursing curricula in the United States is correct to care for women properly. However, other essential SRH content, such as sexual and gender minorities, should be incorporated to ensure that nurses are well trained to meet the SRH needs of all people. The contents they presented were (1) healthcare and human sexuality beyond the focus on reproduction, (2) care options (for example, counseling about pregnancy options), (3) reproductive rights, options, and alternatives during pregnancy, including term management, abortion, fertility control, contraceptives, or the like, (4) postpartum contraception, (5) professional ethics, (6) public health and global health, and (7) quality and safety [11]. 

In Europe, the contents of the SRH subject in undergraduate nursing curricula have traditionally been structured in two areas. One of these areas is Reproductive Health, which includes, one the one hand, topics on the care of women during their pregnancy, childbirth, and postpartum, and on the other hand, topics that focus on the treatment and diagnosis of diseases and conditions that affect the physical and emotional well-being of women. The second area is Sexual Health, which focuses on sexuality and sexual relationships [12,13,14]. In other countries, such as China [15], Ethiopia [16], or Argentina [17], the SRH contents of nursing curricula are mainly focused on reproductive health.

In the case of Spain, a southern European country, to be a nurse, it is necessary to obtain a nursing degree from an official university. This degree means passing a four-year curriculum with 240 ECTS (European Credits Transfer System). Nursing curricula include various subjects, such as Sexual and Reproductive Health. All university curricula are approved by the Ministry of Education and Vocational Training and are periodically reviewed by a specialized ministerial Agency (ANECA) [18]. These reviews are carried out by a minimum group of six recognized university professors who are experts in different areas. This group of experts determines if the contents included in the subjects of each curriculum are adequate to achieve the nursing competencies. To carry out this task, the group of experts does not have a detailed program of the contents that each subject of the curricula must include, but instead, they decide whether the contents of each subject are appropriate or not based on their knowledge and experiences (expert consensus) [19,20]. In practice, this means that universities have a certain autonomy to design and implement the contents of the subjects in the curriculum. The teachers of each subject first decide what content to include, and then the experts review and approve them. Sometimes, experts may request to add or remove some content before being approved. However, since the contents of the subjects are not officially defined, there may be differences in the contents between the approved and revised curricula. Teachers may prefer to include or develop some topics more than others in their subjects. Likewise, it is unknown if there are differences between public and private universities. It has been described that private universities may have a more conservative and timid attitude to address SRH issues than public universities [21].

Another factor that can affect the contents is that the subjects of the nursing curricula do not have the same number of credits in all universities. For example, the SRH subject at the University of Valencia has a study load of 4.5 ECTS [22], and the same subject at the Autonomous University of Barcelona has a load of 6 ECTS [23]. 

Although the curricular contents in SRH necessary for a good nursing performance have not been established in Spain, university professors usually use them as a guide to determine the goals and the objectives of the “Action Plan for Sexual and Reproductive Health: Towards achieving the 2030 Agenda for Sustainable Development in Europe” proposed by the WHO Regional Committee for Europe in 2016 [24], as well as the training recommendations on SRH established in the Sexual and Reproductive Health Strategy of the Ministry of Health of the Government of Spain in 2011 [25].

Regarding Sexual Health, since the WHO established the need to train all nurses in this field, the curricula have gradually included some content on sexuality [26]. Most of the curricula are limited to information on reproductive aspects, contraceptives, and the prevention of sexually transmitted infections [25]. Regarding Reproductive Health, nursing curricula have traditionally included content on pregnancy, childbirth, and the puerperium [25]. However, since the Strategy of Attention to Normal Delivery of the National Health System of Spain in 2007, considered as one of its strategic lines training concerning this process of labor and delivery, the curricula have been updated, deepening the understanding of the reproductive process and the physiology of childbirth from a gender perspective and bearing in mind that it is about attention to a healthy process and not to disease [27].

Given the manifest importance of receiving adequate essential content in SRH during undergraduate nursing studies to acquire basic knowledge and given the ignorance of the exact contents taught in the SRH subject in Spanish universities, a revision of the academic guides of Spanish universities was required performed.

This study aimed to determine which SRH contents are undergraduate nursing students currently receiving in Spanish universities. The main objective was to examine the contents of the SRH subject of the undergraduate nursing curricula of all Spanish universities. Likewise, it was established as a secondary objective to identify differences in SRH contents between public and private universities.

## 2. Materials and Methods

### 2.1. Settings

In Spain, according to data from the Ministry of Universities, in 2020, there were 80 official nursing curricula in force, 55 from public universities and 25 from private universities [28].

Each Spanish university has a public document that contains all the “teaching guides” of the subjects composing the undergraduate nursing curriculum [18,29]. These teaching guides include the contents (topics) taught in each subject during undergraduate training. This study focused exclusively on the contents of the Sexual and Reproductive Health subject.

### 2.2. Design and Sample

A descriptive cross-sectional study was designed. All the teaching guides of the SRH subject of the undergraduate nursing curricula available online were included (*n* = 77, 96.3% of them). The contents (topics) of this subject were reviewed. This study did not require ethical committee approval because it did not involve human participants.

### 2.3. Data Collection

In the first place, the websites of all the Spanish universities that taught undergraduate nursing studies during the 2020/2021 academic year were accessed, and the type of university (public or private) was determined. In the second place, the nursing curricula of these universities were searched, and the year of publication of each one was determined. Next, the teaching guide for the SRH subject of each curriculum was located. Finally, the contents section of each teaching guide was revised.

The teaching guides’ contents (topics) were classified into two areas: Reproductive Health and Sexual Health. Likewise, the following information was also extracted from the teaching guides: number of ECTS (European Credit Transfer System) of the SRH subject, number of hours of face-to-face interaction, the year in which the SRH contents are taught (first, second, third, or fourth), sex of the headteacher of the subject (male or female), and the profession of the headteacher (nurse, midwife, or physician).

### 2.4. Data Analysis

All analyses were derived from quantifying the frequency of appearance (prevalence) of the Sexual and Reproductive Health contents in the teaching guides. Likewise, inferential analyses (test χ^2^) were carried out to determine differences in the contents between public and private universities, respectively. Cramer’s V was also calculated to measure the effect size. The level of significance was set at 0.05. All analyses were carried out with SPSS version 23.

## 3. Results

A total of 77 (96.3%) curricula were included, 52 (65%) from public universities and 25 (31.3%) from private universities. Three curricula were excluded because the teaching guides of the SRH subject were not available online. Most of the curricula were approved between 2016 and 2020 (32, 41.6%), followed by those approved between 2011 and 2015 (29, 37.7%) and finally between 2005 and 2010 (16, 20.8%). 

Regarding the teaching guides of the Sexual and Reproductive Health subject included in the nursing curricula, it was found that they were taught mainly in the third year (45, 58.4%) and second year (29, 37.7%). The headteacher of the SRH subject, in most cases, was a woman (63, 81.8%). Likewise, most headteachers were nurses (41, 53.2%) or midwives (30, 39.0%). A low percentage of headteachers were doctors (6, 7.8%). Sexual and Reproductive Health subjects had a load of 6 ECTS credits in 47 (61%) universities and a median of 36 h of face-to-face master class (min. 3–max. 120).

### Contents in Sexual and Reproductive Health

In the Reproductive Health area, 15 topics were found (Table 1). The most prevalent topic of this area was “Pregnancy”, which was present in 65 teaching guides (84.4%). In contrast, the topic with the lowest prevalence was “Women’s Health Services Administration”, which was included in only three teaching guides (3.9%). Statistically significant differences were found between public and private universities for the topics “Health Problems During Pregnancy” (χ^2^ = 16.347, *p* = 0.038), “Delivery” (χ^2^ = 18.622, *p* = 0.017), “Complications in Labor and Delivery” (χ^2^ = 25.057, *p* = 0.002), “Health Problems in the Puerperium” (χ^2^ = 17.249, *p* = 0.028), and “Women’s Health Education and Research” (χ^2^ = 15.890, *p* = 0.044).

The contents of each topic (subtopics) of the Reproductive Health area are presented in an attached document due to their length (Appendix A). The most frequent subtopics were Anatomy of the Female Reproductive Organs (60, 77.9%), Physiology of the Female Reproductive System (61, 79.2%), pregnancy (65, 84.4%), Bleeding Problems (63, 81.8%), Hypertensive states (64, 83.1%), Infectious Problems (60, 77.9%), Labor and Delivery (61, 79.2%), Puerperium (65, 84.4%), and Climacteric and Menopause (58, 75.3%).

In the Sexual Health area, 14 topics were found (Table 1). The most prevalent topic of this area was “General Concepts”, which was present in 38 teaching guides (49.4%). The topic with the lowest prevalence was “Drugs and sexual behavior”, included in six teaching guides (7.8%). Statistically significant differences were found between public and private universities for the topics “Socio-Anthropology of Sexuality” (χ^2^ = 31.440, *p* = 0.012), “Components of Sexuality” (χ^2^ = 35.952, *p* = 0.003), “Sexual Psychophysiology” (χ^2^ = 32.874, *p* = 0.008), and “Sex Education” (χ^2^ = 20.279, *p* = 0.009).

## 4. Discussion

In this study, all the contents (topics) presented in the teaching guides of the SRH subject of the undergraduate nursing curricula of Spanish universities have been analyzed. The findings show that these contents focus on essential SRH topics (Table 1). 

Most of the fifteen Reproductive Health topics found have a medium-high prevalence in most universities, especially pregnancy, childbirth and postpartum, contraceptive/family planning methods, and sexually transmitted diseases. These results are consistent with those of other previous studies [5,11]. This basic knowledge coverage has been considered sufficient for developing adequate competencies in this area in the US [30].

However, the fourteen Sexual Health topics found are not very frequent; they are taught in few universities. The sexual health contents selected by the Spanish universities ranged from highest to lowest prevalence: concepts related to sex and sexuality, human sexual response (physiology and sexual dysfunctions), sexuality by stages of life, and the physiological, psychological, and sociological factors that affect sexual health. However, topics such as sexual health education from a nursing approach, sexual rights, non-normal behaviors related to sexuality, sexual abuse, and sexual violence are hardly addressed. Likewise, sexuality and sexual health problems in LGBT+ people and sexuality and sexual health problems in cancer patients, for example, are not included as currently recommended [11,24,25]. 

This low prevalence of sexual health content in most curricula does not seem adequate for acquiring the basic knowledge that allows students to develop good competencies in sexual health [21,31]. With little training and content, nursing students may have little knowledge of sexual health and may not provide adequate advice on sexual health as professionals a posteriori [21]. The gaps in Sexual Health education in Spain are similar to those described in 2016 by Aaberg for the USA. This author identified that the content of sexuality was inferior, nonspecific, and limited to general topics, i.e., anatomy and physiology, conception, contraception, normal sexual function, sexual dysfunction, and sexually transmitted infections [32]. Studies are needed to assess the level of knowledge in sexual health of nursing students.

The most common barriers described justifying this low inclusion of Sexual Health topics have been the perception that they are not a curricular priority, time constraints, the religious affiliation of the university, faculty beliefs, and teachers’ lack of qualification or comfort [11]. In general, universities have been reported to have a conservative and timid attitude when it comes to addressing issues of sexuality, especially in private universities [21]. This attitude could explain the differences found for specific contents (such as contraception, assisted reproduction, and sexuality) in this study.

In this study, we have not analyzed whether there are differences in the contents of the most recent SRH didactic guides (2016–2020) compared to the oldest (2005–2010) or among the most recent exclusively. More research is required to analyze the most current guides’ content and show the evolution of SRH content over time. The differences in contents of the didactic guides has not been analyzed depending on the region where the university is located or the ECTS credits of the subject. It would be time to analyze these questions in future studies. It would also be interesting to carry out an exploratory study with the teachers of the SRH subject to determine what content they consider a priority and the barriers they encounter when developing and teaching it in the classroom.

Currently, the legislation for developing undergraduate nursing curricula in Spain does not establish guidelines for the contents, so both the universities and the professors decide on these contents [19]. This could generate differences in the content between the curricula, which can cause inequalities in the training that students receive depending on the university in which they study. In general, the contents of SRH are considered specialized knowledge [20]. Most teachers decide these contents, basically thinking about which ones will benefit the students the most [5]. To select these contents, they usually use as a guide the goals and objectives of the “Action Plan for Sexual and Reproductive Health: Towards achieving the 2030 Agenda for Sustainable Development in Europe” proposed by the WHO Regional Committee for Europe in 2016 [24], as well as the training recommendations on SRH established in the Sexual and Reproductive Health Strategy of the Ministry of Health of the Government of Spain in 2011 [25]. These documents consider it a priority that health sciences’ students receive comprehensive and adequate training in sexual and reproductive health that takes into account at least the following aspects: (1) Comprehensive sexual and reproductive health from a human rights perspective, including intimate partner violence and non-partner sexual violence and exploitation. (2) Socioeconomic, cultural, and gender determinants in SRH and different contexts of vulnerability in different population groups; maternal and perinatal mortality and morbidity. (3) Gender perspective in SRH. Sexual diversity, reality, and needs are related to the sexual health of all people, regardless of their sexual options and orientations and gender identities. Communication in SRH with women and families. (4) Contraception, sexually transmitted infections (STIs), and HIV. (5) Infertility, reproductive cancers, and other benign pathologies of the reproductive system. (6) Sexual and reproductive health at all stages of life, including especially adolescents. (7) Sexual and reproductive health services available. National SRH strategies and programs. (8) The reproductive process: pregnancy, childbirth, and the puerperium (including neonate). Voluntary interruption of pregnancy (Organic Law 2/2010). (9) The process of labor and delivery, based on respect for women and physiology (best clinical practices).

For all this, it is necessary to urgently address the SRH training of undergraduate nursing students in Spain to ensure that all students receive a homogeneous content in SRH that adequately covers their academic and clinical needs. This change does not mean offering a set number of topics, but ensuring that the core content is taught to all students. It is about seeking particular equity in the content and avoiding deficit situations [33]. Likewise, knowing the contents that are currently being studied is essential to carry out a process of improving training in this field. This research can be an essential tool for professors and universities to review the curricula and reflect on the content they teach in Sexual and Reproductive Health subjects. This research can serve as the basis for an update of the teaching guides.

The main limitation of this research is the possibility that some teaching guides have not published all the contents or have presented them in a very general way without specifying them. Perhaps we have underestimated the scope of the SRH content taught in nursing education. However, examining the SRH teaching guides of Spanish universities has made it possible to map what content is taught and which are the most frequent. In addition, in this study, only the teaching guides for the SRH subject were reviewed. It is possible that other subjects of the curricula include some SRH content. For example, the Geriatrics subject may include topics on sexual health at this stage of life [29].

Nurses who receive extensive and accurate training in SRH are likely to provide higher quality care in this field. For this reason, it is crucial to establish a good education in sexual and reproductive health in Spanish universities. The priority strategy in nursing should be to train students so that they have sufficient SRH knowledge, skills, and competencies, and that they can, thus, carry out good care for all people and communities in SRH.

## 5. Conclusions

The contents of SRH in nursing curricula are basic and limited. There is considerable variation in the amount and thematic content among universities. The training focuses primarily on reproductive health. Training in sexual health is the least developed. 

It is necessary to carry out a curricular review and agree on the content of SRH to cover the basic knowledge that nurses need to provide quality care to the entire population.

## Figures and Tables

**Table 1 ijerph-18-11472-t001:** Contents of the teaching guides of the Sexual and Reproductive Health subject in the undergraduate nursing curricula of the Spanish Universities.

	Total Universities (*n* = 77)		Public Universities (*n* = 52)		Private Universities (*n* = 25)				
	n	%	n	%	n	%	Test χ^2^	*p*	Cramer’s V *
Reproductive Health									
1. Human reproduction	61	79.2	42	80.8	19	76.0	28.235	0.703	0.024
2. Human prenatal development	50	64.9	35	67.3	15	60.0	17.347	0.578	0.037
3. Pregnancy	65	84.4	45	86.5	20	80.0	15.896	0.635	0.033
4. Health problems during pregnancy	64	83.1	45	86.5	19	76.0	16.347	**0.038**	0.053
5. Delivery	61	79.2	39	75.0	22	88.0	18.622	**0.017**	0.066
6. Complications in labor and delivery	54	70.1	33	63.5	21	84.0	25.057	**0.002**	0.104
7. Puerperium and lactation	65	84.4	45	86.5	20	80.0	27.129	0.638	0.033
8. Health problems in the puerperium	49	63.6	35	67.3	14	56.0	17.249	**0.028**	0.057
9. Women’s reproductive health care	58	75.3	38	73.1	20	80.0	9.695	0.060	0.035
10. Care for women with reproductive health problems	51	66.2	34	65.4	17	68.0	29.791	0.728	0.013
11. Epidemiology and demography in reproductive health	9	11.7	6	11.5	3	12.0	33.153	0.942	0.003
12. Socio-anthropological aspects in reproductive health	10	13.0	8	15.4	2	8.0	16.249	0.558	0.038
13. Women’s Health Education and Research	30	39.0	22	42.3	8	32.0	15.890	**0.044**	0.052
14. Legislation and Ethics	5	6.5	4	7.7	1	4.0	17.546	0.715	0.019
15. Women’s Health Services Administration	3	3.9	2	3.8	1	4.0	44.813	0.953	0.001
Sexual Health									
1. General concepts	38	49.4	26	50.0	12	48.0	20.108	0.780	0.010
2. Socio-anthropology of sexuality	13	16.9	11	21.2	2	8.0	31.440	**0.012**	0.067
3. Components of sexuality	14	18.2	12	23.1	2	8.0	35.952	**0.003**	0.077
4. Sexual psychophysiology	17	22.1	14	26.9	3	12.0	32.874	**0.008**	0.076
5. Sexuality in puberty and adolescence	21	27.3	13	25.0	8	32.0	15.622	0.584	0.036
6. Sexuality in adulthood	12	15.6	8	15.4	4	16.0	56.542	0.925	0.003
7. Sexuality in pregnancy and the puerperium	12	15.6	9	17.3	3	12.0	16.745	0.681	0.027
8. Sexuality in the climacteric	18	23.4	11	21.2	7	28.0	26.820	0.062	0.034
9. Sexual dysfunctions	18	23.4	12	23.1	6	24.0	27.425	0.910	0.005
10. Health and sexual behavior problems	21	27.3	14	26.9	7	28.0	29.810	0.808	0.006
11. Drugs and sexual behavior	6	7.8	3	5.8	3	12.0	16.562	0.666	0.031
12. Sex education	26	33.8	20	38.5	6	24.0	20.279	**0.009**	0.074
13. Sex and power	11	14.3	7	13.5	4	16.0	18.929	0.759	0.013
14. Approach to sexuality in clinical practice	8	10.4	5	9.6	3	12.0	31.499	0.761	0.012

* Interpretation of Cramer’s V: >0.25 (very strong), >0.15 (strong), >0.10 (moderate), >0.05 (weak), >0 (no or very weak), Bold was used to identify statistically significant differences.

## Data Availability

Data sharing not applicable.

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
