# Peer review of "Contents of the Sexual and Reproductive Health Subject in the Undergraduate Nursing Curricula of Spanish Universities: A Cross-Sectional Study"

_ijerph, 2021, doi:10.3390/ijerph182111472_

Round 1

Reviewer 1 Report

    The introduction does not clearly describe the objectives of the study. There is a lack of a position that justifies what knowledge is necessary in SRH for a good nursing performance and from there to describe what the situation is and to be able to make a proposal for improvement.
2.    In the section on material and method, the content of what is indicated is not well understood, is it assumed that the syllabus is the material of the work? If so, it should be worded better.
3.    Design and sample: It is not specified how many university guides have been used. 
4.    Discussion: It remains very ambiguous and in fact contradicts somewhat with the contents of the introduction. There is a lack of a theoretical framework to justify the need to integrate specific sexual health knowledge and to better describe what kind of knowledge is missing and needs to be integrated.

Author Response

RESPONSE TO REVIEWER COMMENTS NUMBER 1

  1. The introduction does not clearly describe the objectives of the study. There is a lack of a position that justifies what knowledge is necessary in SRH for a good nursing performance and from there to describe what the situation is and to be able to make a proposal for improvement.

Thank you very much for your appreciation.

As far as we know, the curricular contents necessary in SRH on the excellent performance of nurses have not been exhaustively described in the literature. In paragraphs 2, 3, and 4, SRH contents considered appropriate in the US have been included, and paragraph 5 presents contents considered as appropriate in other contexts. Likewise, we have included two paragraphs at the end of the introduction in which we describe the situation in Spain.

  1. In the section on material and method, the content of what is indicated is not well understood, is it assumed that the syllabus is the material of the work? If so, it should be worded better.

Thank you for your valuable comments. Changes to the Methods and Discussion sections have been performed to clarify this issue.

  1. Design and sample: It is not specified how many university guides have been used.

Data about the number of guides reviewed have been included in the methods section.

  1. Discussion: It remains very ambiguous and in fact contradicts somewhat with the contents of the introduction. There is a lack of a theoretical framework to justify the need to integrate specific sexual health knowledge and to better describe what kind of knowledge is missing and needs to be integrated.

Thanks so much again for these comments; they have been very engaging in improving the manuscript. The discussion has been restructured to clarify it. We have removed the contradiction with the content of the introduction by discussing sexual and reproductive health content in the curricula. Contents of the "Action Plan for Sexual and Reproductive Health: Towards achieving the 2030 Agenda for Sustainable Development in Europe" proposed by the WHO Regional Committee for Europe in 2016, as well as the Sexual and Reproductive Health Strategy of the Ministry of Health of the Government of Spain in 2011 have been included as a theoretical basis to justify the need to integrate specific knowledge in SRH. From the results found and these two documents, we have described what kind of knowledge is missing and should be integrated.

Reviewer 2 Report

The chi-squared test is used to compare the distribution of a categorical variable in a sample or a group with the distribution in another one. As the significant test does not tell us the degree of effect, displaying effect size is helpful to show the magnitude of effect. Different measures are needed to achieve this:  Phi (φ), Cramer's V (V)…

The values in the tables are disorganized.

This study has tried to determine if there were differences in the contents between public and private universities respectively.

I encourage the authors to submit it for publication in higher education journals.

Thank you very much. 

Author Response

  1. The chi-squared test is used to compare the distribution of a categorical variable in a sample or a group with the distribution in another one. As the significant test does not tell us the degree of effect, displaying effect size is helpful to show the magnitude of effect. Different measures are needed to achieve this: Phi (φ), Cramer's V (V)…

Thank you very much for your recommendation. We have included the measure: Cramer's V to show the effect size in Table 1.

  1. The values in the tables are disorganized.

We apologize for this error. Tables have been reorganized to improve their presentation and the clarity of the data presented.

  1. This study has tried to determine if there were differences in the contents between public and private universities respectively. I encourage the authors to submit it for publication in higher education journals.

Thank you very much for your comments, as they have helped us improve the manuscript.

Reviewer 3 Report

The article portrays a current topic that is very pertinent and of great interest to public health.

The article is well-structured, has an important purpose.

Some recommendations to the discussion:

Spain does not establish guidelines for the contents, but what are the recommendations of the Spain nursing associations? Or other European countries nursing associations?

From the European concerns some topics could be included:

https://www.euro.who.int/en/health-topics/Life-stages/sexual-and-reproductive-health/publications/2016/action-plan-for-sexual-and-reproductive-health-towards-achieving-the-2030-agenda-for-sustainable-development-in-europe-leaving-no-one-behind-2016

https://doi.org/10.3389/fpubh.2021.656454

Are there differences between the more recent curricula?

Recommendations for the future should be detailed.

The conclusion should more be concrete.

Author Response

  1. The article portrays a current topic that is very pertinent and of great interest to public health. The article is well-structured, has an important purpose.

Thank you very much for your consideration of our work.

  1. Some recommendations to the discussion: Spain does not establish guidelines for the contents, but what are the recommendations of the Spain nursing associations? Or other European countries nursing associations? From the European concerns some topics could be included: https://www.euro.who.int/en/health-topics/Life-stages/sexual-and-reproductive-health/publications/2016/action-plan-for-sexual-and-reproductive-health-towards-achieving-the-2030-agenda-for-sustainable-development-in-europe-leaving-no-one-behind-2016 https://doi.org/10.3389/fpubh.2021.656454

The "Action Plan for Sexual and Reproductive Health: Towards achieving the 2030 Agenda for Sustainable Development in Europe" proposed by the WHO Regional Committee for Europe in 2016, as well as the Sexual and Reproductive Health Strategy of the Ministry of Health of the Government of Spain in 2011 have been considered as the theoretical basis to justify the need to integrate specific knowledge in SRH. From the results found and these two documents, we have described what knowledge is missing and should be integrated. Thank you very much for your recommendation.

  1. Are there differences between the more recent curricula?

Thanks so much for your question. In this study, we have not analyzed whether there are differences in the contents of the most recent SRH teaching guides. A qualitative content analysis on this question and the evolution of SRH content in the study plans of Spanish universities would be fascinating. We have described this limitation in the discussion (lines 261-267)

  1. Recommendations for the future should be detailed.

Based on the results found, it has been indicated the need to carry out an in-depth curricular review of the SRH subject and the need to agree on the SRH content so that all students receive adequate basic training that covers the needs of future nurses. Thank you very much for your recommendation.

  1. The conclusion should more be concrete.

Thank you very much for your comments. We consider that they have been essential to improving the manuscript. We have modified the text of the conclusion to specify it and to answer more explicitly to the proposed objective.

Reviewer 4 Report

The authors evaluate the sexual and reproductive health training of Spanish nursing students through the contents of the sexual and reproductive health subject in the undergraduate nursing curricula. This is a  relevant and interesting topic.

Keywords: The keywords focus on the main dimensions, but could add to the title, including: Comprehensive Sexuality Education and Gender Equality and Sexual Rights.

Introduction:

The rationale for the study was well explained. However, a more comprehensive and more critical overview of global health approaches to sexual and reproductive health diversity of needs across life course and populations would be expected in the introduction. Why the focus only on women’s health?

Your introduction would benefit from presenting your operation framework of sexual and reproductive health and it should be inclusive of the needs of women and girls, men and boys, children (including adolescents), LGBTI people, older men and women, disabled people, those belonging to national or ethnic groups, and religious and linguistic minorities.

Results:  Results focus mainly on the comparison between public and private universities. This should be listed as an objective, presented in the introduction (the relevance of the comparison), and also could be included in the title. What about other possible differences, such as regional, number of ECTS, year of the discipline?

Discussion: A more deep and critical discussion about what specific content would need to be included in nursing teaching programs seems to be missing.

The authors could better detail the study limitations and discuss their implications for the results. An exploratory study with the coordinators of nursing undergraduate courses about the place in the curricula of sexual and reproductive health issues for example would add strength to your results.

Conclusion: Although results are exploratory, implications of the results should be enlightened with key messages for nursing schools, nursing professionals, and for future investigations.

Author Response

  1. The authors evaluate the sexual and reproductive health training of Spanish nursing students through the contents of the sexual and reproductive health subject in the undergraduate nursing curricula. This is a relevant and interesting topic.

Thank you very much for your comment.

  1. Keywords: The keywords focus on the main dimensions, but could add to the title, including: Comprehensive Sexuality Education and Gender Equality and Sexual Rights.

We have included the keywords. Thanks.

  1. Introduction:

The rationale for the study was well explained. However, a more comprehensive and more critical overview of global health approaches to sexual and reproductive health diversity of needs across life course and populations would be expected in the introduction. Why the focus only on women's health?

Your introduction would benefit from presenting your operation framework of sexual and reproductive health and it should be inclusive of the needs of women and girls, men and boys, children (including adolescents), LGBTI people, older men and women, disabled people, those belonging to national or ethnic groups, and religious and linguistic minorities.

Thank you very much for your recommendation. We have integrated the first paragraph to describe sexual and reproductive health approaches, and the second paragraph has been modified to reflect the need to include all people.

  1. Results: Results focus mainly on the comparison between public and private universities. This should be listed as an objective, presented in the introduction (the relevance of the comparison), and also could be included in the title. What about other possible differences, such as regional, number of ECTS, year of the discipline?

The trade-off between public and private universities has been included in the objective and in the introduction. In this study, we have not analyzed whether there are differences in the teaching guides' contents depending on the country's region, the number of ECTS, or the year of publication of the study plan. A qualitative content analysis on these issues would be fascinating. We have described this limitation in the discussion (lines 261-267) and the need to carry out studies in this regard. Thank you very much.

  1. Discussion: A deeper and critical discussion about what specific content would need to be included in nursing teaching programs seems to be missing.

An in-depth and critical review of the discussion has been carried out, including a recommendation on the contents based on the "Action Plan for Sexual and Reproductive Health: Towards achieving the 2030 Agenda for Sustainable Development in Europe" proposed by the WHO Regional Committee for Europe in 2016, as well as the Sexual and Reproductive Health Strategy of the Ministry of Health of the Government of Spain of 2011. Thank you very much for your recommendation.

  1. The authors could better detail the study limitations and discuss their implications for the results. An exploratory study with the coordinators of nursing undergraduate courses about the place in the curricula of sexual and reproductive health issues for example would add strength to your results.

Throughout the discussion, we have included and clarified the limitations of the study and the implications of the results (the need for a curricular revision in SRH). We take note of your recommendation and will carry out shortly an exploratory study with the coordinators of the SSR subject of the nursing degree of the Spanish universities to know, based on their experience, what are the contents that they consider a priority, as well as to determine the barriers they encounter in their development (lines 267-270). Thank you very much.

  1. Conclusion: Although results are exploratory, implications of the results should be enlightened with key messages for nursing schools, nursing professionals, and for future investigations.

We have modified the text of the conclusion to try to clarify and present the conclusion more accurately (lines 322-327). Thank you very much for all your comments.

Round 2

Reviewer 1 Report

The article has been substantially improved and I consider it recommended for publication.

Author Response

a

Reviewer 4 Report

Thank you for the revision of the manuscript. The authors have put a lot of effort into addressing the comments. I believe the manuscript is much clearer now.

I would suggest a stronger debut sentence for your abstract that would highlight the unique role of the nurses in addressing SRH needs for all people at all ages. Also, please revise Appendix A to improve the presentation and the clarity of the data presented.

Author Response

b